# A Gibbs Posterior Framework for Fair Clustering

**DOI:** 10.3390/e26010063

**Published:** 2024-01-11

**Authors:** Abhisek Chakraborty, Anirban Bhattacharya, Debdeep Pati

**Affiliations:** Department of Statistics, Texas A&M University, College Station, TX 77843, USA; anirbanb@stat.tamu.edu (A.B.); debdeep@stat.tamu.edu (D.P.)

**Keywords:** algorithmic fairness, balance, generalized Bayes, minimum cost flow, optimal transport

## Abstract

The rise of machine learning-driven decision-making has sparked a growing emphasis on algorithmic fairness. Within the realm of clustering, the notion of *balance* is utilized as a criterion for attaining *fairness*, which characterizes a clustering mechanism as fair when the resulting clusters maintain a consistent proportion of observations representing individuals from distinct groups delineated by *protected attributes*. Building on this idea, the literature has rapidly incorporated a myriad of extensions, devising fair versions of the existing frequentist clustering algorithms, e.g., k-means, k-medioids, etc., that aim at minimizing specific loss functions. These approaches lack uncertainty quantification associated with the optimal clustering configuration and only provide clustering boundaries without quantifying the probabilities associated with each observation belonging to the different clusters. In this article, we intend to offer a novel probabilistic formulation of the fair clustering problem that facilitates valid uncertainty quantification even under mild model misspecifications, without incurring substantial computational overhead. Mixture model-based fair clustering frameworks facilitate automatic uncertainty quantification, but tend to showcase brittleness under model misspecification and involve significant computational challenges. To circumnavigate such issues, we propose a generalized Bayesian fair clustering framework that inherently enjoys decision-theoretic interpretation. Moreover, we devise efficient computational algorithms that crucially leverage techniques from the existing literature on optimal transport and clustering based on loss functions. The gain from the proposed technology is showcased via numerical experiments and real data examples.

## 1. Introduction

Fairness in algorithmic decision-making aims to mitigate discrimination involving the unfavorable treatment of individuals based on their membership to specific demographic sub-groups identified by *protected attributes*. These protected attributes may encompass factors such as gender, race, marital status, etc., depending on the specific context, and are often delineated by local, national, or international legal frameworks. For instance, in the context of bank loan approvals, the protected attribute *marital status* might encompass labels such as married, divorced, and unmarried applicants. Perhaps unsurprisingly, early research in fairness on machine learning exclusively focused on supervised learning problems. However, there was a genuine need to understand fairness in unsupervised learning settings, especially in clustering problems.

In a seminal work, Chierichetti et al. [1] introduced the concept of *balance* as a criterion for achieving fairness in clustering, defining clustering mechanisms as fair when resulting clusters maintain a common ratio of observations representing individuals from different groups identified by protected attributes. This notion was explored in both the *k*-center and the *k*-median problems, particularly in the *two-color* case. Subsequent articles extended this framework to the more complex *multi-color* cases, addressing the situation where the protected attributes have multiple labels [2]. Esmaeili et al. [3] considered the case with imperfect knowledge of group membership through probabilistic assignments. Bera et al. [4] expanded the scope by enabling users to specify parameters controlling extent of balance in the clusters, considering the general $L_{p}$ objective for the clustering and scenarios where individuals could belong to multiple protected groups. Fairness mechanisms have been explored in different clustering frameworks, such as spectral clustering [5], correlation clustering [6], and hierarchical clustering [7]. Additionally, researchers have investigated the notions of individual fairness [8,9,10] and proportional fairness [11] in the context of clustering. Fairness in clustering has also been studied in combination with other critical aspects of modern machine learning, including privacy [12] and robustness [13]. For a comprehensive review, interested readers are directed to the website on https://www.fairclustering.com/ (accessed on 30 November 2023).

While the follow-up works [2,3,12,14] greatly increased the scope of fair clustering from various aspects, uncertainty quantification associated with the optimal clustering configuration was largely illusive until recently. Chakraborty et al. [15], complementing the existing literature on fair clustering which is almost exclusively based on optimizing appropriate objective functions, took a fully Bayesian approach to tackle the problem of clustering under balance constraints to provide valid uncertainty quantification, developed a concrete notion of optimal recovery in this problem, and devised a scheme for principled performance evaluation of algorithms. In this article, we propose an alternative *generalized Bayesian* fair clustering framework, embedding common clustering loss functions at the heart of the likelihood formulation. Generalized Bayesian methodologies, as exemplified in various works [16,17], are gaining prominence due to their ability to alleviate the necessity of explicitly specifying the complete data generative mechanism. This characteristic allows us to circumnavigate the challenges related to the lack of robustness of fully Bayesian clustering approaches. Moreover, a principled selection of the *temperature parameter* in the generalized likelihood guarantees its adherence to valid decision-theoretic justifications. Finally, ardent care is exercised to devise efficient computational algorithms to carry out posterior inference, crucially leveraging techniques from the existing literature on clustering based on loss functions.

Prior to presenting the proposed methodology, we provide a concise overview of pertinent concepts in generalized Bayesian inference and fair clustering, laying the groundwork for subsequent discussions.

### 1.1. Generalized Bayesian Inference through Gibbs Posterior

In Bayesian inference, the need for likelihood specification introduces a notable obstacle in many practical applications, primarily stemming from apprehensions about the potential misspecification of the statistical model [18,19,20]. The non-parametric Bayesian approaches [21,22] enhance the adaptability of such methodologies, bolstering the robustness of the statistical inferences. Nevertheless, evaluating the trade-off between the added complexity in adopting a fully non-parametric approach and the acquired robustness necessitates meticulous consideration tailored to specific applications. Generalized Bayesian inference [16,23,24,25] offers an alternative model-free approach to circumvent the risk of model misspecification bias as well as excessive complexity in problem formulation. We proceed by recording the definition of generalized Bayesian posteriors, followed by a brief overview of notable contributions within this domain.

To that end, let $u={\left(u_{1},\ldots ,u_{N}\right)}^{T}$ be the observed data, θ∈Θ be the parameter of interest, and π(θ) be the prior on θ. Then, the generalized Bayesian posterior is defined as
(1)π(θ∣λ,u)∝π(θ)exp{−λL(θ∣u)},
where λ>0 is a temperature parameter, and L(θ∣u)>0 is a loss function of choice. The posterior in (Equation 1) is often referred to as the *Gibbs posterior*. Standard Bayesian inference is recovered from (Equation 1), when the loss function λL(θ∣u) is a negative log-likelihood. While Gibbs posteriors have found applications in diverse contexts [16,25,26,27,28,29], it is only in recent times that their role in offering a rational update of beliefs, thus qualifying as genuine posterior distributions, has been established [30]. Further, Syring and Martin [31] provided sufficient conditions for establishing concentration rates for Gibbs posteriors under a sub-exponential type loss function. Martin and Syring [17], Holmes and Walker [32] presented significant developments in methodologies for selecting the temperature parameter λ. Owing to such increasing support for generalized Bayesian inference, we propose a Gibbs posterior-based framework for uncertainty quantification in fair clustering. We additionally introduce computationally efficient algorithms for point estimation and associated uncertainty quantification.

### 1.2. Fairness in Clustering

We shall now formally introduce the notion of *balance* in the context of fair clustering. For a positive integer *t*, denote [t]:={1,…,t}. Suppose we observe data ${\left\{\left(x_{i},a_{i}\right)\right\}}_{i=1}^{N}$, where $x_{i}$ denotes the *d*-variate observation for the *i*-th data unit, and $a_{i}$ the label of the protected attribute. For each *a*, let ${\left\{x_{i}^{\left(a\right)}\right\}}_{i=1}^{N_{a}}$ denote the observations corresponding to the *a*-th level of the protected attribute, where $N_{a}=\sum_{i=1}^{N}1\left(a_{i}=a\right)$ and $\sum_{a=1}^{r}N_{a}=N$. The goal of fair clustering is to assign the data points ${\left\{\left(x_{i},a_{i}\right)\right\}}_{i=1}^{N}$ into clusters $C=(C_{1},\ldots ,C_{K})$, ${⋃}_{k=1}^{K}C_{k}=\left[N\right]$, respecting the notion of balance [1].

**Definition** **1****([**[1]**]).** *Given $\left\{\left(x_{i},a_{i}\right)\in X\times \left[2\right],\;i\in \left[N\right]\right\}$ such that $a_{i}=a$ for $i=\sum_{j=1}^{a-1}N_{j}+1,\ldots ,\sum_{j=1}^{a}N_{j}$ where a∈[2] and $N_{0}$ = 0, the balance in $C_{k}$ is defined as*$$\begin{array}{c} Balance\left(C_{k}\right)=\min_{1\leq j_{1}<j_{2}\leq r}\min\left\{\frac{|C_{kj_{1}}|}{|C_{kj_{2}}|},\frac{|C_{kj_{2}}|}{|C_{kj_{1}}|}\right\} \end{array}$$*where $|C_{kj}|$ denotes the number of observations in $C_{k}$ with a=j. The overall balance of the clustering is $Balance\left(C\right)=\min_{k=1,\ldots ,K}Balance\left(C_{k}\right)$. The higher this measure is for a clustering configuration, the fairer the clustering is.*

Given the aforementioned definition of *balance*, Chierichetti et al. [1] introduced the notion of *fairlets* as minimal fair sets that approximately maintain the selected clustering objective. The authors illustrated that addressing any fair clustering problem involves initially obtaining a fairlet decomposition of the data through the solution of a *minimum cost flow* problem. Subsequently, classical clustering algorithms, such as k-means or k-center, can be employed for further processing. As we eluded to earlier, the follow-up literature in fair clustering [2,3,12,14] involves devising fair versions of the existing frequentist clustering algorithms, e.g., k-means, k-medioids, etc., that aim at minimizing specific loss functions. These approaches lack the quantification of uncertainty linked to the optimal fair clustering configuration, i.e., these approaches only provide clustering boundaries but do not quantify the probabilities of each of the observations belonging to the different clusters. While model-based clustering approaches are specifically designed to answer such questions, they routinely fall prey to even minor model misspecification [19,20]. The generalized Bayesian approach provides a convenient middle ground that not only is immune to such minor model misspecifications, but also provides valid uncertainty quantification associated with the clustering. This article introduces a generalized Bayesian fair clustering framework with inherent uncertainty quantification and presents efficient computational algorithms for posterior inference, leveraging techniques from the clustering literature based on loss functions.

The rest of the article is arranged as follows. In Section 2, we introduce the proposed generalized Bayesian fair clustering procedure. Section 3 presents an efficient MCEM and a MCMC scheme to carry out posterior analysis under the proposed framework. Section 4 and Section 5 provide detailed numerical experiments and real data examples to delineate the key gain from the proposed methodology over competing methods. Finally, we conclude with a discussion.

## 2. Methodology

### 2.1. Preliminaries

Chierichetti et al. [1] introduced the notion of *fairlets*, minimal sets that adhere to fair representation while approximately preserving a clustering objective. Given the observed data $\left\{\left(x_{i},a_{i}\right)\in X\times \left[2\right],\;i\in \left[N\right]\right\}$, fair clustering via fairlets [1] involves first decomposing data into a set of *m* fairlets, and calculate the *m* fairlet centers. Let $U(\subset X^{m})$ denote the class of all such “*m* fairlet centers”. Let $L_{f}:U\to R^{+}$ denote loss function utilized for the fairlet decomposition. The optimal fairlet decomposition $u^{★}\in U$ is then expressed as
(2)$$\begin{array}{c} u^{★}=\underset{u\in U}{arg min}L_{f}\left(u\right). \end{array}$$This optimization problem in (Equation 2) is often recognized as the *minimum cost flow* problem [33].

Prior to progressing further, it is imperative to conduct a comprehensive examination of the constituents comprising the loss function $L_{f}$. To that end, suppose the two labels of the protected attribute is represented in a 1:t ratio in the observed data, i.e., $t\times N_{1}=N_{2}$. We wish to find a perfectly balanced clustering of the observed data, i.e., the two labels of the protected attribute should be represented in a 1:t ratio in each of the clusters. The construction of an optimal (1,t)-fairlet decomposition of the observed data involves solution of a constrained binary optimal transport problem. First, we define the $N_{1}\times N_{2}$ cost matrix
$$\begin{array}{c} L=\left(\left(l_{ik}\right)\right)=((D\left(x_{i},x_{N_{1}+k}\right))),\;i\in \left[N_{1}\right],\;j\in \left[N_{2}\right], \end{array}$$
where D(w,v)≥0 quantifies the discrepancy between *w* and v∈X. We introduce a column sum vector $c=t\times 1_{N_{1}}$ and a row sum vector $r=1_{N_{2}}$, where $1_{s}$ is a vector of *s* 1 s. Given the two fixed vectors r,c, we define a polytope of $N_{1}\times N_{2}$ binary matrices
$$\begin{array}{c} U\left(r,c\right):=\left\{B|B1_{N^{\prime }}=r;\;B^{T}1_{N^{\prime }}=c\right\}, \end{array}$$
with fixed margin r,c and solve the constrained binary optimal transport problem [34]
$$\begin{array}{c} B^{\prime }={\mathrm{argmin}}_{B\in U(r,c)}\;\langle B,L\rangle , \end{array}$$
where $\langle B,L\rangle =\mathrm{tr}\left(B^{T}L\right)$. The matrix $B^{\prime }=\left(\left(b_{ik}^{\prime }\right)\right)$ describes an optimal (1,t)-fairlet decomposition. That is, if $b_{ik_{1}}^{\prime }=\ldots =b_{ik_{t}}^{\prime }=1$ for some $i\in [N_{1}]$ and $1\leq k_{1}<\ldots <k_{t}\leq N_{2}$, then ${\left(x_{i},x_{N_{1}+k_{1}},\ldots ,x_{N_{1}+k_{t}}\right)}^{T},\;i\in {\left[N\right]}_{1}$ defines the fairlets. The fairlet centers are represented as $u^{★}={\left(u_{1}^{★},\ldots ,u_{m}^{★}\right)}^{T}$, acquired through averaging observations within the respective fairlets. Finally, we define a map
$$\begin{array}{c} \xi :U(r,c)\to U, \end{array}$$
that takes a binary matrix B with fixed margins (r,c), representing a fairlet decomposition of $\left\{\left(x_{i},a_{i}\right)\in X\times \left[2\right],\;i\in \left[N\right]\right\}$ to *m*-fairlet centers u∈U. Then, loss function $L_{f}$ is represented as
$$\begin{array}{c} L_{f}\left(\xi \left(B\right)\right)=\langle B,L\rangle ,\;B\in U\left(r,c\right). \end{array}$$Efficient off-the-shelf algorithms for optimizing the loss function are routinely available.

Next, given the optimal fairlet decomposition of the observed data $u^{★}={\left(u_{1}^{★},\ldots ,u_{m}^{★}\right)}^{T}\in U$, Chierichetti et al. [1] proposed to invoke existing machinery for traditional clustering algorithms, e.g., k-means, k-center, etc., to cluster the *m*-fairlet centers into *K* groups. We focus on the flexible class of clustering mechanisms characterized by a factorized loss $L_{c}\left(C|u^{★}\right)$. Throughout the article, we assume that the number of clusters *K* is fixed, e.g., it is known or has been selected in an exploration phase. Let $u_{\left(k\right)}^{★}$ denote the center of the fairlet centers $\left\{u_{i}^{★}:i\in C_{k}\right\}$ belonging to cluster $C_{k}$, for k∈[K]. Then, the clustering is characterized by factorized loss takes the form
(3)$$L_{c}\left(C|u^{★}\right)=\sum_{k=1}^{K}\sum_{i\in C_{k}}D\left(u_{i}^{★},u_{\left(k\right)}^{★}\right),\;C:\left|C\right|=K,$$
where $D(u_{i}^{★},u_{\left(k\right)}^{★})\geq 0$ is a function of $u_{i}^{★}$ and $u_{\left(k\right)}^{★}$ which quantifies the discrepancy of the *i*-th unit from the *k*-th cluster. The formulation in (Equation 3) encapsulates a large class of common clustering costs. For example, suppose we assume that $u_{\left(k\right)}^{★}$ represents the arithmetic means of the vectors $u_{i}^{★},\;i\in C_{K}$ for any k=1,…,K. Then, the k-means loss function takes the form
$$L_{c}\left(C|u^{★}\right)=\sum_{k=1}^{K}\sum_{i\in C_{k}}\left|\right|u_{i}^{★}-u_{\left(k\right)}^{★}{\left|\right|}_{2}^{2}.$$Importantly, efficient off-the-shelf algorithms [35] for solution of the clustering problem in (Equation 3) are routinely available. In summary, Chierichetti et al. [1] critically exploits the existing tools in minimum cost flow problem (Equation 2) and factorized loss-based clustering (Equation 3) to obtain the optimal fair clustering configuration. In a subsequent work, we shall integrate this methodology within the generalized Bayesian inference framework to quantify uncertainty associated with the optimal fair clustering configuration.

### 2.2. Generalized Bayesian Fair Clustering

Given the observed data $\left\{\left(x_{i},a_{i}\right)\in X\times \left[2\right],\;i\in \left[N\right]\right\}$, we recall that $U(\subset X^{m})$ denote the class of all “*m* fairlet centers”. We placed a uniform prior on the space of all possible fairlet decompositions U, i.e., we assume
(4)$$\begin{array}{c} \pi \left(u\right)=\frac{1}{\left|U\right|},\;u\in U. \end{array}$$Our framework is easily modified to consider more elaborate priors, but we focus on the uniform case throughout the paper. Then, the *generalized Bayes posterior* for *fairlet decomposition* takes the form
(5)$$\pi \left(u|\lambda_{f},{\left\{\left(x_{i},a_{i}\right)\right\}}_{i=1}^{N}\right)\;\propto \;\frac{\exp\left\{-\lambda_{f}L_{f}\left(u\right)\right\}}{\sum_{u\in U}\exp\left\{-\lambda_{f}L_{f}\left(u\right)\right\}},$$
where $\lambda_{f}$ is a temperature parameter.

Given a fairlet decomposition u∈U, which may be different from the optimal fairlet decomposition $u^{★}$, a typical Bayesian model for clustering [36,37,38] is based on the assumption that observations follows from
(6)$$\left(u_{i}|\theta_{k},i\in C_{k}\right)\overset{\mathrm{ind}}{∼}\pi \left(u_{i}|\theta_{k}\right),\;k\in \left[K\right],$$
where $\theta_{k}\overset{\mathrm{iid}}{∼}\pi \left(\theta \right)$ for k∈[K]. Under the above model and prior specification, the posterior distributions of clustering configurations take the form
(7)$$\pi \left(C|u\right)\;\propto \;\pi \left(C\right)\prod_{k=1}^{K}\left[\int_{\Theta }\prod_{i\in C_{k}}\pi \left(u_{i}|\theta \right)\pi \left(\theta \right)d\theta \right],$$
where π(C) is the prior probability of C, π(u∣θ) is the within-cluster likelihood, and π(θ) is the prior distribution on the cluster-specific parameters. While Equation (Equation 7) serves as the foundation for an extensive body of literature on Bayesian clustering, it gives rise to significant practical challenges. The integral often does not admit a closed form expression, introducing computational complexities. Furthermore, the posterior of clustering configurations is highly sensitive to the precise specifications of data generating mechanism π(u∣θ). Such model-based clustering frameworks (Equation 6) are routinely criticized for various aspects. Firstly, clustering may just serve as a convenient preprocessing step, and there might not be distinct groups present in the data. Moreover, even if such groups exist, the distribution of the data within each cluster is unlikely to precisely adhere to the chosen distribution π(u∣θ). In terms of computational aspects, even when the data accurately conform to the selected mixture distribution, we frequently encounter substantial computational bottlenecks, especially in high dimensions [39,40,41].

To address these brittleness issues, Rigon et al. [29] implemented a Gibbs posterior framework tailored for clustering, aiming to navigate around these challenges. Let the loss function L(C∣u) be as in (Equation 3). In this article, our attention is directed towards employing uniform clustering priors of the form
(8)$$\pi \left(C\right)=\frac{1}{S(n,K)},\;C:\left|C\right|=K,$$
where $S\left(n,K\right)=1/K!\sum_{k=0}^{K}{\left(-1\right)}^{K-k}K!{\left\{\left(K-k\right)!k!\right\}}^{-1}k^{n}$ is the Stirling number of the second kind. Prior (Equation 8) is uniform over partitions having *K* components. Although the framework readily accommodates more intricate clustering priors, our emphasis in this paper remains on the uniform case. The generalized Bayes posterior under a *generalized Bayes product partition model* has the form
$$\pi \left(C|\lambda_{c},u\right)\;\propto \;\prod_{k=1}^{K}\exp\left\{-\lambda_{c}\sum_{i\in C_{k}}\;D\left(u_{i},u_{\left(k\right)}\right)\right\},\;C:\left|C\right|=K,$$
where $\lambda_{c}$ is a temperature parameter and $\left\{u_{\left(k\right)},\;k\in \left[K\right]\right\}$ are the *K* cluster centers. We have now assembled all the necessary components to introduce a generalized Bayesian posterior framework for the fair clustering problem.

In this article, we still focus on the uniform clustering prior in (Equation 8), and utilize the uniform prior on the space of all possible fairlet decompositions U, introduced in (Equation 4). Then, the *generalized Bayes posterior* for *fair clustering* takes the form
$$\begin{array}{cc} \pi (C,u|\left(\lambda_{f},\lambda_{c}\right),{\left\{\left(x_{i},a_{i}\right)\right\}}_{i=1}^{N})\;\; & \propto \;\frac{\exp\left\{-\lambda_{f}L_{f}\left(u\right)\right\}}{\sum_{u\in U}\exp\left\{-\lambda_{f}L_{f}\left(u\right)\right\}}\times \exp\left\{-\lambda_{c}L_{c}\left(C|u\right)\right\}. \end{array}$$Under the assumption of factorised clustering loss, the posterior simplifies to
(9)$$\begin{array}{c} \pi \left(C,u|\left(\lambda_{f},\lambda_{c}\right),{\left\{\left(x_{i},a_{i}\right)\right\}}_{i=1}^{N}\right)\;\propto \;\frac{\exp\left\{-\lambda_{f}L_{f}\left(u\right)\right\}}{\sum_{u\in U}\exp\left\{-\lambda_{f}L_{f}\left(u\right)\right\}}\times \prod_{k=1}^{K}\exp\left\{-\lambda_{c}\sum_{i\in C_{k}}\;D\left(u_{i},u_{\left(k\right)}\right)\right\}, \end{array}$$
such that C:|C|=K. We employ the methodology proposed in [32] for the selection of the temperature parameters $\left(\lambda_{f},\lambda_{c}\right)$.

A natural competitor for the proposed methodology based on the joint posterior in (Equation 9) is the fair clustering with fairlets [1]. Subsequent to [1], many follow up articles proposed suggestions for improved computational scalability of the fair clustering task [42,43]. However, from a methodological perspective, [1] still serves as the primary go-to method.

**Proposition** **1.**
*Let $\pi (C,u|\left(\lambda_{f},\lambda_{c}\right),{\left\{\left(x_{i},a_{i}\right)\right\}}_{i=1}^{N})$ denote the joint posterior of the fairlet decompositions of the observed data and clustering configurations in *(*Equation 9*)*. Then, we denote the posterior mode by*

$$\begin{array}{c} \left(C^{\mathrm{MAP}},u^{\mathrm{MAP}}\right)=\underset{C,u}{arg max}\left[\pi (C,u|\left(\lambda_{f},\lambda_{c}\right),{\left\{\left(x_{i},a_{i}\right)\right\}}_{i=1}^{N})\right]. \end{array}$$

*The clustering configuration $C^{\mathrm{MAP}}$ does not coincide with the optimal clustering obtained via fair clustering via fairlets [1].*


**Proof.** From Equation (Equation 9), we note that the quantities $\left(C^{\mathrm{MAP}},u^{\mathrm{MAP}}\right)$ are computed via solving
$$\begin{array}{c} \underset{C,u}{arg max}\left[\pi \left(C,u|\left(\lambda_{f},\lambda_{c}\right),{\left\{\left(x_{i},a_{i}\right)\right\}}_{i=1}^{N}\right)\right]=\underset{C,u}{arg min}\left[\lambda_{f}L_{f}\left(u\right)+\lambda_{c}\sum_{k=1}^{K}\sum_{i\in C_{k}}\;D\left(u_{i},u_{\left(k\right)}\right)\right], \end{array}$$
where $\lambda_{f}\geq 0$ and $\lambda_{c}\geq 0$. On the other hand, the optimal clustering obtained via fair clustering via fairlets is calculated in a two-step process: first, the optimal fairlet decomposition $u^{★}\in U$ is obtained by
$$\begin{array}{c} u^{★}=\underset{u\in U}{arg min}L_{f}\left(u\right), \end{array}$$
and then, the optimal clustering is obtained by minimizing the factorized loss given $u^{★}$,
$$C^{\mathrm{FCF}}=\underset{C}{arg min}L_{c}\left(C|u^{★}\right)=\underset{C}{arg min}\sum_{k=1}^{K}\sum_{i\in C_{k}}D\left(u_{i}^{★},u_{\left(k\right)}^{★}\right),\;C:\left|C\right|=K.$$This completes the proof.    □

In Proposition 1, we prove that the maximum a posteriori fair clustering configuration obtained via maximizing the joint posterior in (Equation 9) is different from the optimal clustering configuration obtained using fair clustering via fairlets. For the purpose of point estimation, we argue that $C^{\mathrm{MAP}}$ should be preferred over $C^{\mathrm{FCF}}$, since it is obtained via maximizing the posterior with respect to the fairlet decomposition and clustering configuration simultaneously. The $C^{\mathrm{FCF}}$ estimator, on the other hand, should be considered as an approximation of our $C^{\mathrm{MAP}}$ estimator since $C^{\mathrm{FCF}}$ is practically obtained via first solving
$$\begin{array}{c} u^{★}=\underset{u\in U}{arg max}\left[\exp\left\{-\lambda_{f}L_{f}\left(u\right)\right\}\right], \end{array}$$
then, given the optimal fairlet decomposition $u^{★}$ of the observed data, solving
$$\begin{array}{c} C^{\mathrm{FCF}}=\underset{C}{arg max}\left[\prod_{k=1}^{K}\exp\{-\lambda_{c}\sum_{i\in C_{k}}\;D\left(u_{i}^{★},u_{\left(k\right)}^{★}\right)\}\right]. \end{array}$$In numerical studies, we observed that the maximum a posteriori estimate of the proposed fair k-means clustering via Gibbs posterior turns out to be very similar to the fair clustering with fairlets estimator (refer to Figure 1 and Figure 2). The proposed Gibbs posterior-based approach, unlike a traditional optimization-based approach [3], also provides uncertainty quantification associated with the clustering configurations along with the maximum a posteriori estimate.

We conclude this section by discussing some other salient features of the posterior in (Equation 9). Firstly, sampling in the space of all possible fairlet decompositions, rather than only focusing on the optimal fairlet decomposition, enables us to take into account the uncertainties arising from this step. This, in turn, crucially enables us to conduct joint inference on (C,u). Secondly, the suggested formulation for clustering the fairlet centers avoids specifying an underlying data generation mechanism and adeptly circumvents the computation of integrals in high-dimensional spaces. Finally, the assumed factorized loss for the clustering significantly simplifies the posterior computation (refer to Section 3 for details).

As we eluded to earlier, ardent care is required to ensure efficient sampling from the joint posterior of fairlet decompositions and clustering configurations in (Equation 9). To that end, we first develop an intermediate Monte Carlo expectation maximization (MCEM) scheme, followed by a subsequent full Markov Chain Monte Carlo (MCMC) scheme to sample from the posterior.

## 3. Posterior Analysis

### 3.1. Sampling Scheme

We develop a Gibbs sampling scheme to sample from the joint posterior of fairlet decompositions and clustering configurations in (Equation 9). We achieve this via iterating over a step-by-step scheme, as per the common practice. We first draw a potentially non-optimal fairlet decomposition of the data and then sample the clustering indices given the specific fairlet decomposition of the data. We cycle through the steps until convergence.

We now put our computational strategy in concrete terms. Suppose the two labels of the protected attribute are represented in a 1:t ratio in the observed data, where *t* is an integer and assume $t\times N_{1}=N_{2}$. Our goal is to explore the space of the perfectly balanced clustering configurations, compute the maximum a posteriori perfectly balanced clustering configuration, and quantify the uncertainty attached to it. This task can be accomplished via iterating over two steps: **Step 1.** sampling in the space of the all possible (1,t)-fairlet decomposition of the observed data; and **Step 2.** given a potentially non-optimal (1,t)-fairlet decomposition of the observed data, sample in the space of all possible clustering configurations of the fairlet centers. We cycle through **Step 1** and **Step 2** until convergence.

**Step** **1** **(Sampling** **the** **Fairlets).**Sampling in the space of the all possible (1,t)-fairlet decomposition of the observed data is further carried out in two steps. In **(i)**, we simply obtain the optimal (1,t)-fairlet decomposition of the observed data. However, to accomplish our goal of quantifying the uncertainty associated with the fair clustering, we first need to take into account the uncertainty associated with the (1,t)-fairlet decomposition of the observed data. To that end, in step **(ii)**, we utilize an innovative Metropolis step to explore other potentially non-optimal (1,t)-fairlet decompositions *near* the optimal (1,t)-fairlet decomposition of the observed data. The two steps follow.

**(i)** We demonstrate how we can utilize discrete optimal transport to obtain the optimal (1,t)-fairlet decomposition of the observed data. We undertake the following steps. First, we define the $N_{1}\times N_{2}$ cost matrix
$$\begin{array}{c} L=\left(\left(l_{ik}\right)\right)=((D\left(x_{i},x_{N_{1}+k}\right))),\;i\in \left[N_{1}\right],\;j\in \left[N_{2}\right], \end{array}$$
column sum vector $c=1_{N_{1}}$, and row sum vectors $r=1_{N_{2}}$, where $1_{s}$ is a vector of *s* 1 s. Next, given the two vectors r,c, we define the polytope of $N_{1}\times N_{2}$ binary matrices
$$\begin{array}{c} U\left(r,c\right):=\left\{B|B1_{N^{\prime }}=r;\;B^{T}1_{N^{\prime }}=c\right\} \end{array}$$
and solve the constrained binary optimal transport problem [34]
(10)$$\begin{array}{c} B^{\prime }={\mathrm{argmin}}_{B\in U(r,c)}\;\langle B,L\rangle , \end{array}$$
where $\langle B,L\rangle =\mathrm{tr}\left(B^{T}L\right)$. The matrix $B^{\prime }=\left(\left(b_{ik}^{\prime }\right)\right)$ describes an optimal (1,t)-fairlet decomposition. That is, for any $i\in [N_{1}]$, if $b_{ik_{1}}^{\prime }=\ldots =b_{ik_{t}}^{\prime }=1$ for some $1\leq k_{i,1}<\ldots <k_{i,t}\leq N_{2}$, then
$$\begin{array}{c} \left\{{\left(x_{i},x_{N_{1}+k_{i,1}},\ldots ,x_{N_{1}+k_{i,t}}\right)}^{T},\;i\in \left[N_{1}\right]\right\} \end{array}$$
defines a fairlet decomposition of the observed data.

**(ii)** We shall see that a weighted rectangular loop [15] update $B^{\prime \prime }=\left(\left(b_{ik}^{\prime \prime }\right)\right)$ on the $B^{\prime }$ matrix provides an alternative, but potentially non-optimal (1,t)-fairlet decomposition of the observed data. Then, for any $i\in [N_{1}]$, if $b_{ik_{1}}^{\prime \prime }=\ldots =b_{ik_{t}}^{\prime \prime }=1$ for some $1\leq k_{i,1}<\ldots <k_{i,t}\leq N_{2}$, then $\left\{{\left(x_{i},x_{N_{1}+k_{i,1}},\ldots ,x_{N_{1}+k_{i,t}}\right)}^{T},\;i\in \left[N_{1}\right]\right\}$ defines a fairlet decomposition of the observed data.

To describe the weighted rectangular loop scheme, let us denote a non-negative weight matrix representing the relative probability of observing a count of 1 at the (i,j)-th cell as $\Omega =\left(\omega_{ij}\right)=\left(\left(\exp-\lambda_{c}l_{ik}\right)\right)\in \left[0,\infty \right)$. Then, the likelihood associated with the observed binary matrix H∈U(r,c) is
$$\begin{array}{c} P\left(H\right)=\left(1/\kappa \right)\prod_{i,j}\omega_{ij}^{h_{ij}},\;\kappa =\sum_{H\in U(r,c)}\;\prod_{i,j}\omega_{ij}^{h_{ij}}. \end{array}$$Let $U^{\prime }\left(r,c\right)=\left\{H\in U\left(r,c\right):\;P\left(H\right)>0\right\}$ denote the subset of matrices in U(r,c) with positive probability. Then, for $H_{1},H_{2}\in U^{\prime }\left(r,c\right)$, the relative probability of the two observed matrices is
(11)$$\begin{array}{c} \frac{P(H_{1})}{P(H_{2})}=\frac{\prod_{\left\{i,j:h_{1,ij}=1,h_{2,ij}=0\right\}}\omega_{ij}^{h_{1,ij}}}{\prod_{\left\{i,j:h_{1,ij}=0,h_{2,ij}=1\right\}}\;\omega_{ij}^{h_{2,ij}}}. \end{array}$$With these notations, we are all set to introduce a *weighted rectangular loop algorithm* (W-RLA) for non-uniform sampling from the space of fixed margin binary matrices H∈U(r,c), given the weight matrix $\Omega =\left(\omega_{ij}\right)\in \left[0,\infty \right)$. To that end, let us first record that the identity matrix of order 2, the 2×2 matrix with all zero diagonal entries, and all one off-diagonal entries are referred to as *checker-board* matrices. W-RLA is then described in Algorithm 1 in complete generality. The validity of the W-RLA scheme is established in Chakraborty et al. [15].
**Algorithm 1:** Weighted rectangular loop algorithm [15]
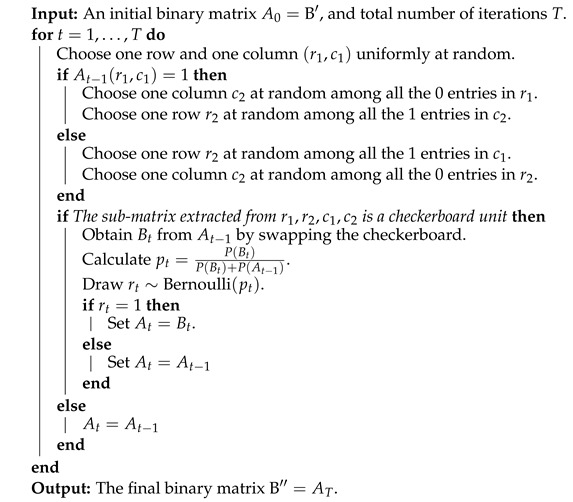


**Step** **2** **(Sampling** **Clustering** **Indices).**The fairlet decomposition of the observed data,
$$\begin{array}{c} \left\{{\left(x_{i},x_{N_{1}+k_{i,1}},\ldots ,x_{N_{1}+k_{i,t}}\right)}^{T},\;i\in \left[N_{1}\right]\right\}, \end{array}$$
induced by the binary matrix $B^{\prime \prime }$, is summarized by the set of fairlet centers $u={\left(u_{1},\ldots ,u_{m}\right)}^{T}\in U$, acquired through averaging observations within the respective fairlets. Given a set of fairlet centers $u={\left(u_{1},\ldots ,u_{m}\right)}^{T}\in U$, we adopt a set of strategies developed in Rigon et al. [29] for sampling the clustering indices. Implementation is generally simpler and more efficient compared to mixture models. A straightforward Gibbs sampler implementation showcases favorable mixing properties.

Here and elsewhere, we refer to each of the fairlet centers as units. Suppose $c_{-i}=\left(c_{1},\ldots ,c_{i-1},c_{i+1},\ldots ,c_{n}\right)$ denotes the set of clustering indices without the *i*-th unit, and let $\left\{C_{1,-i},\ldots ,C_{K,-i}\right\}$ be the induced partition of the fairlet centers. Suppose $u_{(k),-i}$ denotes the center of the units $\left\{u_{i}:i\in C_{k,-i}\right\}$. In Gibbs sampling we cyclically reallocate the indicators $c_{i}$ by sampling from their full conditionals. Then, the conditional distribution of $c_{i}$ given $c_{-i}$ is
(12)$$\begin{array}{c} P\left(c_{i}=k|c_{-i},\lambda_{f},u\right)\;\propto \;\exp\left\{-\lambda_{f}\left[\sum_{i^{\prime }\in C_{k}}D\left(u_{i^{\prime }},u_{\left(k\right)}\right)-\sum_{i^{\prime }\in C_{k,-i}}D\left(u_{i^{\prime }},u_{(k),-i}\right)\right]\right\}, \end{array}$$
for k=1,…,K and for any partition C:|C|=K.

In summary, an **MC-EM** algorithm to sample from the posterior in Equation (Equation 9) involves first finding the optimal fairlet decomposition via (Equation 2) and then sampling the clustering indices via the scheme described in (Equation 12). A complete **MCMC** scheme to sample from the posterior needs a bit more work. First, given the observed data $\left\{\left(x_{i},a_{i}\right)\in X\times \left[2\right],\;i\in \left[N\right]\right\}$, the optimal fairlet decomposition is obtained via the optimization problem in (Equation 10). We propose a non-optimal fairlet decomposition of the data-weighted rectangular loop scheme in Algorithm 1. For given fairlet decomposition, sample the clustering indices via the scheme described in (Equation 12). We repeat the steps until convergence.

The proposed Metropolis within Gibbs sampling algorithm automatically enable us to obtain asymptotically exact samples from the target joint posterior [44], and hence, we prefer it over common optimization-based algorithms, such as variational inference [45], which does not enjoy such guarantees and often provides inadequate uncertainty quantification. Further, while gradient-based MCMC algorithms for discrete spaces [46,47] are becoming increasingly popular in the literature, development of of such schemes under the proposed setup involve significant work, since exploring the space of all possible fairlet decompositions require sampling in constrained spaces.

### 3.2. Posterior Summaries

Suppose $S_{T}=\left\{s_{1},\ldots ,s_{T}\right\}$ is *T* post burn-in draws from the marginal posterior of clustering distribution. For each clustering configuration $s\in S_{T}$, one can obtain the association matrix η(s) of order N×N, whose (i,j)-th element is an indicator whether the *i*-th and the *j*-th observation are clustered together or not. Element-wise averaging of these *T* association matrices yields the pairwise clustering probability matrix, denoted by $\bar{\eta }=\frac{1}{T}\sum_{s\in S_{T}}\eta \left(s\right)$. To summarize the MCMC draws, we adopt the least square model-based clustering introduced in Dahl [48] to obtain $s_{\mathrm{LS}}={\mathrm{argmin}}_{s\in S_{T}}\sum_{i=1}^{N}\sum_{j=1}^{N}{\left(\eta_{ij}\left(s\right)-{\bar{\eta }}_{ij}\right)}^{2}.$ Since $s_{\mathrm{LS}}\in S_{T}$ by construction, the notion of balance is retained in the resulting $s_{\mathrm{LS}}$.

We can also the obtain misclassification probabilities $1-{\bar{\eta }}_{ii^{★}}$ of the observations, where $i^{★}$ denotes the medioid of the cluster to which the *i*-th observation was allocated in the maximum a posteriori (MAP) clustering. The quantity $1-{\bar{\eta }}_{ii^{★}}$ approximates the probability that the *i*-th unit is allocated to a cluster different from the cluster in the MAP clustering.

### 3.3. Hyperparameter Tuning

We shall now delve into the considerations regarding the selection of the number of clusters *K* and the temperature parameter $\lambda ={\left(\lambda_{f},\lambda_{c}\right)}^{T}$ within the joint posterior of clustering indices and fairlet decompositions, as expressed in Equation (Equation 9). Firstly, aligning with the approach presented in [29], we conceive the number of clusters denoted as *K*, not as an intrinsic attribute of the data to be estimated but rather as a parameter reflecting the desired level of granularity for partitioning observations. Hence, we propose the subjective specification of *K* as intrinsic to the specified loss function, rather than inferring it from the data. Alternatively, during an exploratory phase, conventional techniques such as the “elbow” rule may be employed to determine a suitable *K*.

It is important to recall that a significant advantage of mixture model-based Bayesian clustering lies in its capacity to automatically deduce the optimal number of clusters (*K*) from the available data. Nonetheless, akin to various model-based methodologies, model-based clustering is susceptible to even minor misspecifications in the component specific parametric distributions. Consequently, the derived estimates for the number of clusters (*K*) through these procedures exhibit inconsistency [19,20]. Conversely, a notable disadvantage inherent in Gibbs posterior-based clustering approaches, including the methodology introduced in this article, is the imperative requirement to fix the number of clusters, designated as *K*, before initiating the clustering procedure. In developing a Gibbs posterior-based clustering framework with an undetermined number of clusters, the complexities escalate concerning the solicitation of loss, the selection of an appropriate temperature parameter, etc. The direct application of generalized Bayesian clustering with a variable *K* may result in undesirable properties in the clustering posterior [29]. This presents a challenging yet captivating avenue for prospective investigation.

Next, our attention shifts to the calibration of the temperature parameters $\lambda ={\left(\lambda_{f},\lambda_{c}\right)}^{T}$. Determining the parameter $\lambda_{f}$ within the Gibbs posterior of the fairlet decomposition in (Equation 5) is executed by employing the general principles discussed in Holmes and Walker [32] for assigning a value to a power likelihood in generalized Bayesian models. With regard to tuning $\lambda_{c}$ in the Gibbs posterior associated with the clustering loss given a specific fairlet decomposition u∈U, we note that [29] highlighted the association between the generalized Bayes product partition model and mixtures of exponential dispersion models. These models represent a generalization of regular exponential families, characterized by a *dispersion parameter*. It was further illustrated in [29] that $\lambda_{c}$ in the definition of the loss function coincides with this dispersion parameter. This alternative probabilistic representation aids in interpreting the loss and facilitates the elicitation of $\lambda_{c}$, an otherwise challenging task as acknowledged in the literature [30,32]. We are now equipped with all of the necessary components to demonstrate the efficacy of the proposed methodology through numerical experiments and real data examples.

Noteworthy, the proposed fair clustering approach and computational strategies work for any factorized clustering loss, e.g., Minkowsski’s loss, Bregman k-means loss, etc. For the sake of demonstration, we present Gibbs posterior with k-means clustering loss in the following numerical experiments and real data examples.

## 4. Experiments

### 4.1. Well-Specified Case

In this simulation, we consider a generating mechanism such that there are two distinct groups present in the data. Moreover, each of the attribute specific components precisely follows an isotropic bivariate normal distribution. Specifically, we consider the following setup. In the first cluster, 20 individuals with a=1 are generated from $N_{2}\left(\mu_{11},S\right)$ and 30 individuals with a=2 are generated from $N_{2}\left(\mu_{21},S\right)$. In the second cluster, 30 individuals with a=1 are generated from $N_{2}\left(\mu_{12},S\right)$, and 20 individuals with a=2 are generated from $N_{2}\left(\mu_{22},S\right)$, where $\mu_{11}={\left(4,4\right)}^{\prime },\;\mu_{21}={\left(2,2\right)}^{\prime },\;\mu_{12}={\left(10,10\right)}^{\prime },\;\mu_{22}={\left(8,8\right)}^{\prime }$, and $S=4I_{2}$ where $I_{2}$ is the two-dimensional identity matrix. This data generation mechanism ensures that individuals with a=1 and a=2 are equally represented in the observed sample. The goal is to obtain two (i.e., we assume K=2 is known) completely balanced clusters along with uncertainty quantification associated with the clustering indices.

The maximum a posteriori estimate of the vanilla k-means clustering via Gibbs posterior [29] in Figure 1 coincides with the k-means estimator and provides clusters with balance 0.62. The maximum a posteriori estimate of the proposed fair k-means clustering via Gibbs posterior in Figure 1 turns out to be very similar to the fair clustering with fairlets estimator [1] and provides clusters with balance 1. The Gibbs posterior-based approaches, unlike the optimization-based approaches Table 1, also provide uncertainty quantification associated with the clustering configurations along with the maximum a posteriori estimate.

### 4.2. Misspecified Case

In this simulation, we consider a generating mechanism such that there are two distinct groups present in the data. Moreover, each of the attribute specific components does not follow an isotropic bivariate normal distribution. Specifically, we consider the following setup. In the first cluster, 20 individuals with a=1 are generated from $t_{2}\left(\mu_{11},S\right)$ and 30 individuals with a=2 are generated from $t_{2}\left(\mu_{21},S\right)$. In the second cluster, 30 individuals with a=1 are generated from $t_{2}\left(\mu_{12},S\right)$ and 20 individuals with a=2 are generated from $t_{2}\left(\mu_{22},S\right)$, where $\mu_{11}={\left(4,4\right)}^{\prime },\;\mu_{21}={\left(2,2\right)}^{\prime },\;\mu_{12}={\left(10,10\right)}^{\prime },\;\mu_{22}={\left(8,8\right)}^{\prime }$, $S=3I_{2}$ where $I_{2}$ is the two-dimensional identity matrix, $t_{\nu }\left(\mu ,S\right)$ is the bivariate Student’s t-distribution with mean μ, scale matrix *S*, and degrees of freedom ν. This data generation mechanism ensures that individuals with a=1 and a=2 are equally represented in the observed sample. Note that for ν=2, the variance of the $t_{\nu }\left(\mu ,S\right)$ distribution does not exist, and under this data generation scheme outliers are expected. The goal is to obtain two (i.e., we assume K=2 is known) completely balanced clusters along with uncertainty quantification of the clustering indices.

The maximum a posteriori estimate of the vanilla k-means clustering via the Gibbs posterior [29] in Figure 2 coincides with the k-means estimator and provides clusters with balance 0.67. The maximum a posteriori estimate of the proposed fair k-means clustering via Gibbs posterior in Figure 2 turns out to be very similar to the fair clustering with fairlets estimator [1] and provides clusters with balance 1. The Gibbs posterior-based approaches, unlike the k-means and the fair clustering with fairlets estimators, also provide uncertainty quantification associated with the clustering configurations along with point estimates.

Under the above misspecified data generative mechanism, we further try fair clustering via Gibbs posteriors based on two distinct clustering losses—k-means loss and Manhattan dissimilarities—to compare them with respect to robustness to outliers. It is important to underscore that neither of these clustering costs aligns with the underlying generative process. To ensure the comparability of outcomes, we establish K = 2 for both methodologies. Subsequently, we compute the corresponding co-clustering matrices, which are presented graphically in Figure 3. The visual representation strongly suggests superior performance of the clustering approach based on Manhattan pairwise dissimilarities when contrasted with the one relying on squared Euclidean loss. In the k-means scenario, the presence of outliers unreliable uncertainty quantification. These observations align with expectations, as the historical use of absolute deviations in lieu of squared losses has been a strategy to enhance the robustness of clustering.

## 5. Benchmark Data Sets

We assess the effectiveness of the proposed approach in comparison to established methodologies using well-known benchmark data sets from the UCI repository [49]. These data sets have been previously examined in the fair clustering literature [1,2,3].

### 5.1. Credit Card Data

We opted for numerical attributes, including age and credit limit, to characterize data points within the Euclidean space. Marital status (categorized as married or unmarried) is designated as the sensitive dimension. We conducted a sub-sampling of 120 individuals from the data set, ensuring a target balance of 1. The objective is to achieve three completely balanced clusters (assuming K=3 is known) while simultaneously quantifying uncertainty associated with the clustering indices.

The maximum a posteriori (MAP) estimate for vanilla k-means clustering through a Gibbs posterior aligns with the conventional k-means estimator, yielding clusters with a balance of 0.27. Conversely, the MAP estimate for fair k-means clustering via a Gibbs posterior roughly aligns with the fair clustering with fairlets estimator, resulting in clusters with a balance of 1. Notably, the Gibbs posterior-based approaches in Figure 4, in contrast to both the k-means and fair clustering with fairlets estimators, additionally provide uncertainty quantification pertaining to the clustering configurations alongside MAP estimates.

### 5.2. Diabetes Data

We analyze a data set sourced from the UCI repository consisting of the health outcomes of patients in relation to diabetes. Numerical attributes such as age and time spent in the hospital serve as points in Euclidean space, while gender is designated as the sensitive dimension. Through a sub-sampling process involving 965 individuals from the data set, a gender ratio of 4:5 is maintained, achieving a target balance of 0.8. The objective is to derive four completely balanced clusters (assuming K=4 is known) while concurrently quantifying uncertainty associated with the clustering indices.

The maximum a posteriori (MAP) estimate for vanilla k-means clustering using the Gibbs posterior aligns with the conventional k-means estimator, yielding clusters with a balance of 0.23. In contrast, the MAP estimate for fair k-means clustering through Gibbs posterior roughly aligns with the fair clustering with fairlets estimator, resulting in clusters with a balance of 0.8. Notably, the Gibbs posterior-based approaches in Figure 5, unlike both the k-means and fair clustering with fairlets estimators, additionally furnish uncertainty quantification related to the clustering configurations alongside the MAP estimators.

### 5.3. Portuguese Banking Data

Subsequently, we turn our attention to the Portuguese banking data set, which comprises individual records corresponding to each phone call conducted during a marketing campaign by a Portuguese banking institution. Each record encapsulates information about the client engaged by the institution. For the representation of points in the Euclidean space, we have selected numerical attributes such as age, balance, and duration. Our clustering objective involves achieving balance between married and unmarried clients. Through a sub-sampling process, we reduced the data set to 939 records, maintaining a married-to-not-married client ratio of (2,1), thereby establishing a target balance of 1/2. The primary aim is to derive four completely balanced clusters (assuming K=4 is known), coupled with an assessment of uncertainty in the clustering indices.

The maximum a posteriori (MAP) estimate for vanilla k-means clustering using the Gibbs posterior aligns with the conventional k-means estimator, resulting in clusters with a balance of 0.06. Conversely, the MAP estimate for fair k-means clustering through the Gibbs posterior roughly aligns with the fair clustering with fairlets estimator, yielding clusters with a balance of 1/2. Importantly, the Gibbs posterior-based approaches, in contrast to both the k-means and fair clustering with fairlets estimators, offer additional insight by providing uncertainty quantification associated with the clustering configurations alongside the MAP estimators.

## 6. Discussion

Despite recent advancements [2,3,12,14] that significantly expanded the scope of fair clustering, uncertainty quantification associated with the optimal clustering configuration remained elusive until recently. In a recent contribution, Chakraborty et al. [15] extended the current body of literature on fair clustering by adopting a novel model-based approach to address clustering under balance constraints. Adopting a generative modeling perspective enabled them to offer valid uncertainty quantification linked to the optimal fair clustering configuration. However, fair clustering frameworks based on such naive mixture models often exhibit fragility in the presence of model misspecification and usually entail notable computational challenges. The main contribution of the current article is the proposed generalized Bayesian fair clustering framework, that inherently provides valid uncertainty quantification while avoiding significant complexities in problem formulation. Secondly, we develop efficient computational algorithms for posterior inference, leveraging techniques from the prevailing literature on clustering based on loss functions as well as computational optimal transport.

A significant limitation of Gibbs posterior-based clustering approaches, including the method proposed in this article, lies in the need to fix the number of clusters, denoted as *K*, prior to the clustering process. When considering an indeterminate number of clusters within a Gibbs posterior-based clustering framework, the challenges intensify with regard to loss elicitation, the selection of a suitable temperature parameter, and associated considerations. A straightforward implementation of generalized Bayesian clustering with a variable *K* may lead to undesirable behaviors in the resulting posterior [29]. This provides a challenging yet intriguing avenue for future enquiry.

In conclusion, we reiterate that the generalized Bayesian inference framework presents a potentially advantageous middle ground in trustworthy machine learning applications, bridging the gap between traditional Bayesian inference and inference based on loss functions. This framework allows for valid uncertainty quantification even in the presence of mild model misspecification, all while potentially maintaining computational scalability. Important directions for future investigation involve the development of Gibbs posterior frameworks for various other fair clustering paradigms, including correlation clustering [6], hierarchical clustering [7], functional data clustering [50], etc. The practical significance of generalized Bayesian inference in fair clustering in conjunction with other pivotal facets of modern machine learning such as privacy [12] and robustness [13] is also noteworthy.

## Figures and Tables

**Figure 1 entropy-26-00063-f001:**
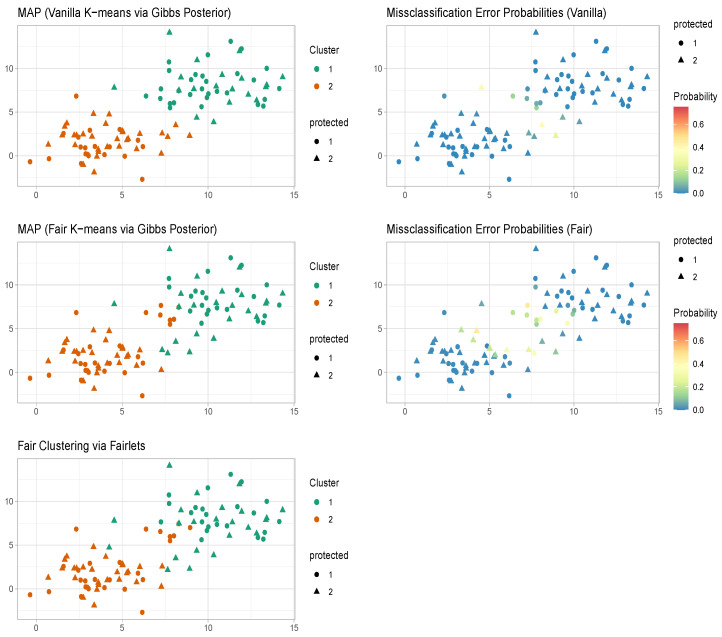
Well-specified case. K-means and fair k-means through Gibbs posterior with K=2. We plot the maximum a posteriori clustering configurations and misclassification error probabilities obtained from the posterior samples, via the scheme in Section 3.2.

**Figure 2 entropy-26-00063-f002:**
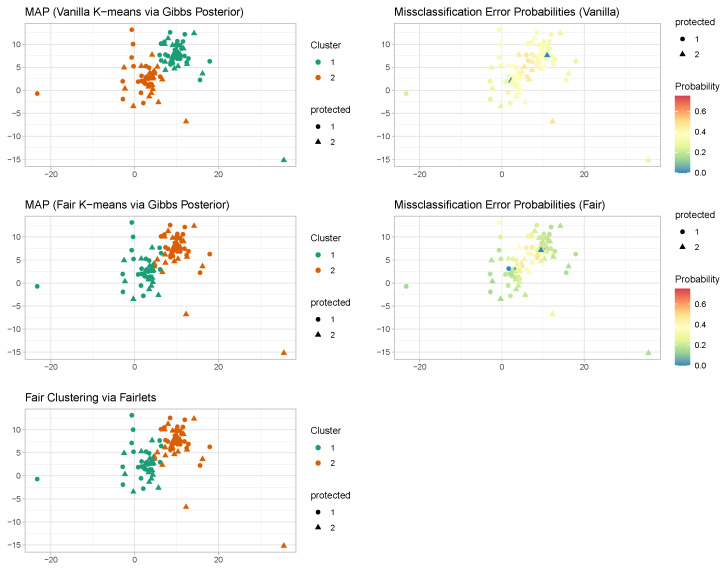
Misspecified case ($L_{2}$ loss). K-means and fair k-means through Gibbs posterior with K=2. We plot the maximum a posteriori clustering configurations and misclassification error probabilities obtained from the posterior samples, via the scheme in Section 3.2.

**Figure 3 entropy-26-00063-f003:**
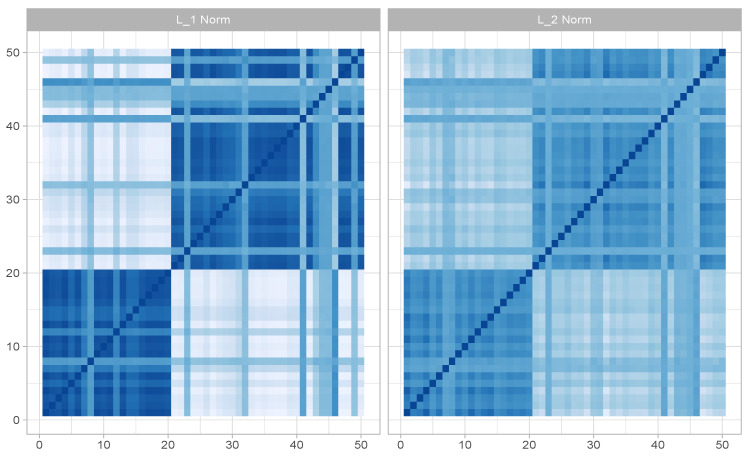
Misspecified case (fair clustering via Gibbs posteriors with $L_{1}$ versus $L_{2}$ loss). Fair k-means through Gibbs posterior with K=2. We plot the co-clustering probability matrix obtained from the posterior samples. The colors are indicative of probabilities, ranging from white indicating low probability to deep blue indicating a high probability.

**Figure 4 entropy-26-00063-f004:**
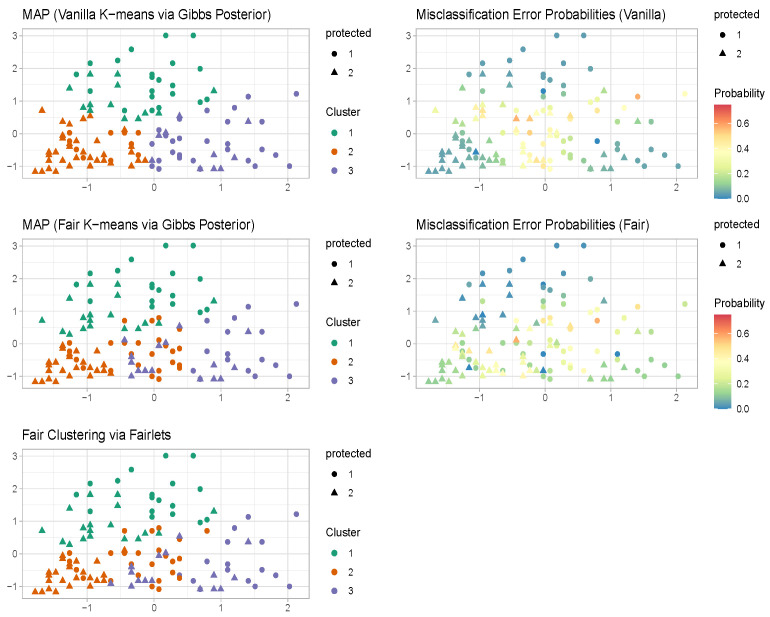
Credit Card Data. K-means and fair k-means through Gibbs posterior with K=3. We plot the maximum a posteriori clustering configurations and misclassification error probabilities obtained from the posterior samples via the scheme in Section 3.2.

**Figure 5 entropy-26-00063-f005:**
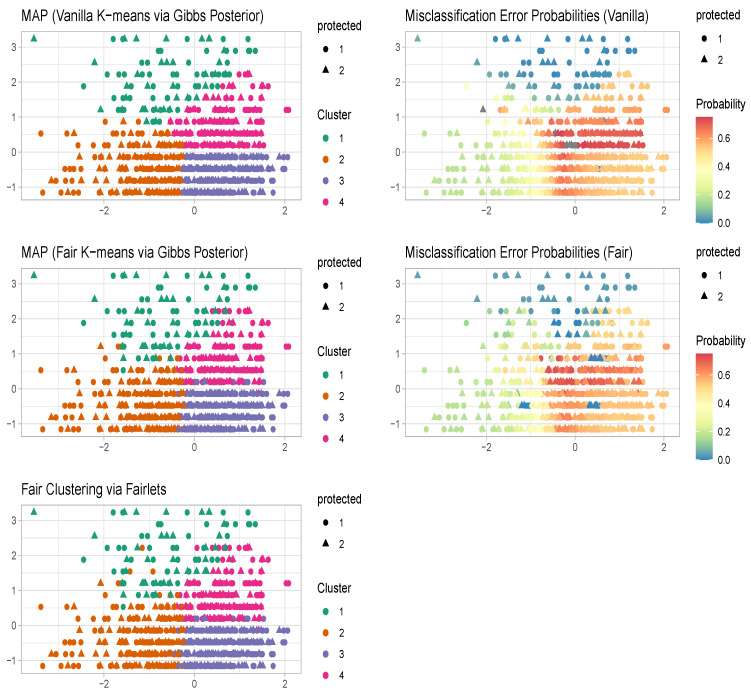
Diabetes Data. Comparison of k-means, fair clustering via fairlets, and fair k-means through a Gibbs posterior with K=4. We plot the maximum a posteriori clustering configurations and misclassification error probabilities obtained from the posterior samples, via the scheme in Section 3.2.

**Table 1 entropy-26-00063-t001:** Overview of Methods.

Method	Fairness	Uncertainty Quantification
K-means	✗	✗
K-means via Gibbs posterior	✗	✓
Fair clustering via fairlets	✓	✗
Fair clustering via Gibbs posterior	✓	✓

## Data Availability

We use well-known benchmark data sets from the UCI repository [49], that have been previously examined in the fair clustering literature.

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
