# Peer review of "A Gibbs Posterior Framework for Fair Clustering"

_entropy, 2024, doi:10.3390/e26010063_

Round 1
Reviewer 1 Report
Comments and Suggestions for Authors
Ensuring fairness in ML models has not received as much attention as it could, both in the deep net community and also across practitioners of computational statistics/ML. This paper helps add to a growing body of work seeking to embed fairness (via some constraints) into well known and widely used algorithms.
The paper is well written, with an adequate (though not extensive) review followed by a presentation of the approach proposed by the authors.
Although the paper is quite well formulated, there are a few areas that could be improved, especially to help readers understand decisions made, especially those that seem to bypass Bayesian approaches that could perhaps have improved the methods even more.
Comments
—------------
Page 2. You introduce Gibbs sampling as an approach to Bayesian Inference. This is presented without much justification, in that there should be reasons why Gibbs is the methodology of choice rather than, eg alternate MCMC approaches or variational Bayes. Bother the latter have been used extensively for full Bayesian mixture modeling.
L87 selecting THE temperature
L98 surely the goal of a Bayesian model is to assign data to degrees of belief regarding their set membership, subject to constraints.
L131 In a Bayesian model we would cast the fairlet decomposition in terms of a PDF over u, not a minimum loss which is just a standard maximum likelihood approach.
L158- surely the benefit of Bayesian clustering or mixture modeling is that K does not need to be determined ahead of time. It is impractical to assume it is known and doing it in a heuristic phase fails to note that K is part of the model so need to be jointly infered along with everything else. This needs some serious justification as I see it as a major limitation of the method.
Section 3. Here it seems like you move to a really Bayesian approach, but there are many ambiguous comments regarding conditioning on an optimized set of fairleta, rather than infering the distribution over them etc. This section is particularly confusing.
Why did you go with sample based approaches then use a poor model for the groupings? K means has well known problems, which at least need discussion. Notably it's sensitive to data scale, so is non-invariant as there is no covariance around the means to compute a Mahanalobis distance. Further, the full mixture model (here a Gaussian mixture) has good Bayesian methods in the literature, placing full Dirichlet Multinomial over the allocations, Wishart on the covariances and Normal on the means. It seems to me that having priors on the Dirichlet could directly enforce the fairness rules, so it's really unclear why this arguably simpler approach was not tried, given it has the benefit of beings fully Bayesian. Can you explain and comment please?
Section 5. It would have been ideal to see direct comparison with existing approaches along with metrics of success, defined and explained. You can add tables of comparison results with clearly justified error or performance metrics.
Reviewer 2 Report
Comments and Suggestions for Authors
Title: A Gibbs Posterior Framework for Fair Clustering
The study discusses the evolution of algorithmic fairness in machine learning, particularly in clustering. It introduces a new probabilistic approach to fair clustering that addresses issues like uncertainty quantification and model misspecification with less computational complexity. The proposed generalized Bayesian framework, supported by efficient computational algorithms using optimal transport and existing clustering techniques, demonstrates its effectiveness through numerical experiments and real data examples. However, the following comments should be addressed before the manuscript can be considered for publication.
1. The abstract needs enhancement, particularly in its structure and content. References are typically not included in an abstract, and the motivation behind the study is not clearly introduced. The current abstract indicates the motivation stems from existing literature focused on optimizing loss functions, which is not a good reason for the research. Instead, the authors should concisely summarize the existing limitations in the field within the abstract, without relying on external references.
2. The introduction of generalized Bayesian Inference through Gibbs Posterior, as presented in line 65, appears to have the same introduction pattern found in “A generalized Bayes framework for probabilistic clustering” on its second page. The authors should include a citation to this original study in this section.
3. What is the purpose of defining the positive integer t in line 94? It seems the following derivation does not use this variable.
4. The authors claimed that accurately measuring the uncertainty associated with achieving the optimal fair clustering configuration is still a largely unresolved challenge. This statement, found in line 114, requires a more detailed explanation. Specifically, the authors should clarify what aspects or factors make the quantification of this uncertainty elusive and why it remains a significant issue in the field.
5. Regarding the proposed method, the authors have not included a comparison with existing methodologies. This omission limits the comprehensiveness of this study, as it becomes challenging to evaluate the efficiency, effectiveness, and accuracy of the new method without a benchmark method for comparison. Such a comparative analysis is crucial for validating the proposed approach.
6. The authors should provide a conclusion section for this study. In conclusion, the authors should summarize their work concisely and point out the results clearly. Also, the authors need to acknowledge any limitations of the proposed method. This will help others better understand the method’s applicability and identify areas where further development and refinement may be needed.
Round 2
Reviewer 1 Report
Comments and Suggestions for Authors
Thank you very much for updating your manuscript in accordance with my review suggestions. I hope the paper is easier for readers to put into context and understand the key contributions.